# A multi-mechanism approach reduces length of stay in the ICU for severe COVID-19 patients

Fernando Valerio Pascua[1☯], Oscar Diaz[2☯], Rina Medina[3☯], Brian Contreras[4☯], Jeff Mistroff[5☯], Daniel Espinosa[6☯], Anupamjeet Sekhon[7☯], Diego Paz Handal[8☯], Estela Pineda[9☯], Miguel Vargas Pineda[10☯], Hector Pineda[11☯], Maribel Diaz[12☯], Anita S. Lewis[13☯], Heike Hesse[14☯], Miriams T. Castro Lainez[15☯], Mark L. Stevens[16☯], Miguel Sierra- Hoffman[17☯], Sidney C. Ontai[18☯], Vincent VanBuren[19]*

1 Department of Critical Care Hospital Leonardo Martínez, San Pedro Sula, Cortés, Honduras, 2 Department of Critical Care Hospital Regional del Norte Instituto Hondureño de Seguridad Social, San Pedro Sula, Cortés, Honduras, 3 Universidad Nacional Autónoma de Honduras, Tegucigalpa, Honduras, 4 Detar Family Medicine Residency Program, Texas A&M College of Medicine Victoria, Victoria, TX, United States of America, 5 Detar Family Medicine Residency Program, Texas A&M College of Medicine Victoria, Victoria, TX, United States of America, 6 Victoria Hospitalist Associates, Detar Hospital, Victoria, TX, United States of America, 7 Detar Family Medicine Residency Program, Texas A&M College of Medicine, Victoria, TX, United States of America, 8 Universidad Nacional Autónoma de Honduras, San Pedro Sula, Cortés, Honduras, 9 Hospital Leonardo Martínez, San Pedro Sula, Cortés, Honduras, 10 Hospital Leonardo Martínez, San Pedro Sula, Cortés, Honduras, 11 Department of Neurology, Hospital Mario Catarino Rivas, San Pedro Sula, Cortés, Honduras, 12 Department of Allergy and Immunology, Instituto Hondureño de Seguridad Social, Sociedad Hondureña de Inmunología, Tegucigalpa, Honduras, 13 Pharmacy Department, El Campo Memorial Hospital, El Campo, TX, United States of America, 14 Neurology Department, Universidad Nacional Autónoma de Honduras, Tegucigalpa, Honduras, 15 Facultad de Ciencias Médicas, Universidad Nacional Autónoma de Honduras, Tegucigalpa, Honduras, 16 Research Department, Texas A&M College of Medicine, Detar Family Medicine Residency Program, Victoria, TX, United States of America, 17 Research and Infectious Disease Department, Texas A&M College of Medicine, Detar Family Medicine Residency Program, Victoria, TX, United States of America, 18 Detar Family Medicine Residency Program Department, Texas A&M University College of Medicine, Victoria, TX, United States of America, 19 Round Rock Clinical Campus, Texas A&M College of Medicine, Round Rock, TX, United States of America

☯ These authors contributed equally to this work.

* vanburen@tamu.edu

**Data Availability Statement:** All relevant data are within the manuscript and its Supporting Information files.

## Abstract

### Purpose

COVID-19 pandemic has multifaceted presentations with rising evidence of immune-mediated mechanisms underplay. We sought to explore the outcomes of severe COVID-19 patients treated with a multi-mechanism approach (MMA) in addition to standard-of-care (SC) versus patients who only received SC treatment.

### Materials and methods

Data were collected retrospectively for patients admitted to the intensive care unit (ICU). This observational cohort study was performed at five institutions, 3 in the United States and 2 in Honduras. Patients were stratified for MMA vs. SC treatment during ICU stay. MMA treatment consists of widely available medications started immediately upon hospitalization. These interventions target immunomodulation, anticoagulation, viral suppression, and oxygenation. Primary outcomes included in-hospital mortality and length of stay (LOS) for the index hospitalization and were measured using logistic regression.

**Funding:** The authors received no specific funding for this work.

**Competing interests:** The authors have declared that no competing interests exist

**Abbreviations:** COVID-19, Coronavirus disease 2019; ARDS, Adult Respiratory Distress Syndrome; MMA, Multi- Mechanism Approach; SC, Standard Care; ICU, Intensive Care Unit; SD, Standard Deviation; IRB, Institutional Review Board; EWS, Early Warning Scale; Q- SOFA, Quick Sequential Organ Failure Assessment score; LOS, Length Of Stay; WHO, World Health Organization; COT, Conventional Oxygen Therapy; NIV, Non-invasive ventilation; HFNC, High Flow Nasal Cannula; FDP, Fibrinogen Degradation Products; DD, D-Dimer; HCQ, Hydroxychloroquine; AZ, Azithromycin; LMWH, Low-Molecular-Weight-Heparin; RCT, Randomized Controlled Trial.

## Results

Of 86 patients admitted, 65 (76%) who had severe COVID-19 were included in the study; 30 (46%) patients were in SC group, compared with 35 (54%) patients treated with MMA group. Twelve (40%) patients in the SC group died, compared with 5 (14%) in the MMA group (p-value = 0.01, Chi squared test). After adjustment for gender, age, treatment group, Q-SOFA score, the MMA group had a mean length of stay 8.15 days, when compared with SC group with 13.55 days. ICU length of stay was reduced by a mean of 5.4 days (adjusted for a mean age of 54 years, p-value 0.03) and up to 9 days (unadjusted for mean age), with no significant reduction in overall adjusted mortality rate, where the strongest predictor of mortality was the use of mechanical ventilation.

## Conclusion

The finding that MMA decreases the average ICU length of stay by 5.4 days and up to 9 days in older patients suggests that implementation of this treatment protocol could allow a health-care system to manage 60% more COVID-19 patients with the same number of ICU beds.

## Introduction

In mid-December of 2019, large clusters of patients presented to local hospitals in Wuhan, China in severe respiratory distress with associated hypoxia and imaging that demonstrated bilateral opacities amongst other findings. However, the novel SARS-CoV-2 virus was not identified as the causative pathogen until early January 2020 [1]. The virus rapidly spread worldwide and was declared a pandemic by the World Health Organization (WHO) on March 11, 2020 [2]. Early in the pandemic, characteristics of the virus were unknown. Therefore, the initial treatment recommendations were made based on other epidemic coronaviruses SARS-CoV-1 and MERS-CoV described in 2003 and 2012, respectively. SARS-CoV-1 and MERS-CoV caused a febrile respiratory disease complicated by Acute Respiratory Distress Syndrome (ARDS), kidney failure and cytokine release syndrome in some cases [3]. In previous studies carried out in patients infected with SARS-CoV-1 and MERS-CoV, no difference was observed in mortality with the use of corticosteroids, which included pooled data from other conditions [3]. Based on this experience, for cases of suspected and/or confirmed COVID 19, WHO classified the disease as mild, moderate, severe, and critical (**Table 1**) in the Clinical Management Guide for Severe Acute Respiratory Infection, as of March 13, 2020 [2]. The WHO advised against empiric antiviral treatment and corticosteroids outside the context of a randomized controlled trial (RCT). These WHO guidelines were followed as the standard of care throughout most countries including the USA and Honduras.

As of March 2020, no therapeutic interventions for severe COVID-19 had been approved by the US Food and Drug Administration or the Honduran Agencia de Regulación Sanitaria. The high death rates associated with COVID-19 prompted the search for an efficacious and affordable therapeutic approach that would be readily available to developed and developing nations alike. A multiple-mechanism approach (MMA) was designed which included widely available medications thought to target early immunomodulation, anticoagulation, and viral suppression to prevent catastrophic cytokine release syndrome and potential progression to respiratory failure, shock, and multi-organ dysfunction (**Fig 1**) [4, 5].

The first component of the MMA addresses immunomodulation and consists of corticosteroids, colchicine, and tocilizumab. Steroids inhibit neutrophil chemotaxis signaling, reducing

**Table 1. Standard of care (SOC) treatment and recommendations.**

| Clinical Classification | Definition | Treatment | Recommendation |
|---|---|---|---|
| Mild | Patients with uncomplicated upper respiratory tract viral infection, may have non-specific symptoms (fever, cough, anorexia, malaise, fatigue, sore throat, headache, etc.) | Symptomatic, such antipyretic for fever | Self-isolation |
| Severe | Fever or suspect viral infection, plus tachypnea and oxygen saturation < 93% on room air | Symptomatic Supplementary oxygen, starting nasal cannula till 5 L/min, if patient continue to have increased work of breathing or hypoxemia used a mask reservoir bag (10–15 L/min) | Hospitalization Target oxygen saturation $\geq$ 93% Conservative fluids if shock is not suspected Empirical antibiotics if Sepsis suspected |
| *ARDS* | Oxygenation impairment: Mild ARDS: 200 mmHg < PaO2/FiO2a$\leq$ 300 mmHg with PEEP or CPAP $\geq$ 5_ cmH2O, or non-ventilated • Moderate ARDS: 100 mmHg < PaO2/FiO2 $\leq$ 200_ mmHg with PEEP $\geq$ 5 cmH2O, or non-ventilated) • Severe ARDS: PaO2/FiO2 $\leq$ 100 mmHg with PEEP $\geq$ 5 cmH2O, or non-ventilated. | Mechanical Ventilation Prone position 12–16 hrs. Deep sedation Neuromuscular blockade, according to clinical condition | Admission to ICU Mechanical ventilation using lower tidal volumes (4-8mL/kg of predicted body weight) and lower inspiratory pressure (plateau pressure <30mmHg) In patients with moderate or severe ARDS higher PEEP Reduced the incidence of venous thromboembolism (use pharmacological prophylaxis of low molecular-weight heparin) |

Based on WHO guidelines. SOC treatment included mechanical ventilation, neuromuscular blockade, self-pronation, intravenous fluids, antibiotics, and vasopressor support.

the production of interleukin (IL)-1, thus reducing neutrophil-platelet interaction and aggregation [6]. Colchicine has been thought to exert its actions by the inhibition of microtubule polymerization and leukocyte infiltration through inhibition of the NLRP3 inflammasome [7]. These features suggest that the simultaneous use of corticosteroids and colchicine could be a safe and synergistic anti-inflammatory therapeutic approach for severe COVID-19 patients. In unpredictable circumstances, patients may overcome the double anti-inflammatory barrier and require a third anti-inflammatory rescue agent such as tocilizumab. This recombinant humanized anti-human IL-6R monoclonal antibody of the IgG1 subtype IL-6 plays a key role in suppressing the COVID-19 related cytokine release syndrome [8].

The second component addresses the hypercoagulable state now known after autopsy findings globally. Another significant cause of death is pulmonary thrombosis including small and medium caliber vessels [9]. Several studies reported that the severity of the disease evolution has a strong positive correlation with elevated coagulation markers, such as D-Dimer (DD) and Fibrinogen degradation products (FDP). Thus, SARS-Cov-2 can activate the coagulation cascade inducing a procoagulant state [10]. There are ongoing discussions on whether heparin and low-molecular-weight-heparin (LMWH) reduce mortality as well as halt progression to the more severe stages of the disease. In addition to its anticoagulant property, heparin also has anti-inflammatory properties which may prove beneficial for this disease process [11].

Nonpharmacological interventions play an integral role in the MMA approach. High flow oxygenation and self-proning are effective interventions to maximize gas exchange and reduce the occurrence for invasive mechanical ventilation [12, 13]. Due to the unpredictable course of the disease, timing is essential in the implementation of the MMA approach, which may decrease mortality and hospital length of stay.

## Methods

We conducted a retrospective cohort study, data was collected from medical records in different dates according to Institutions Review Boards (IRB) permission, between June 10[th] to August 6[th], 2020, including five different hospitals: 1. Hospital Regional del Norte, Instituto

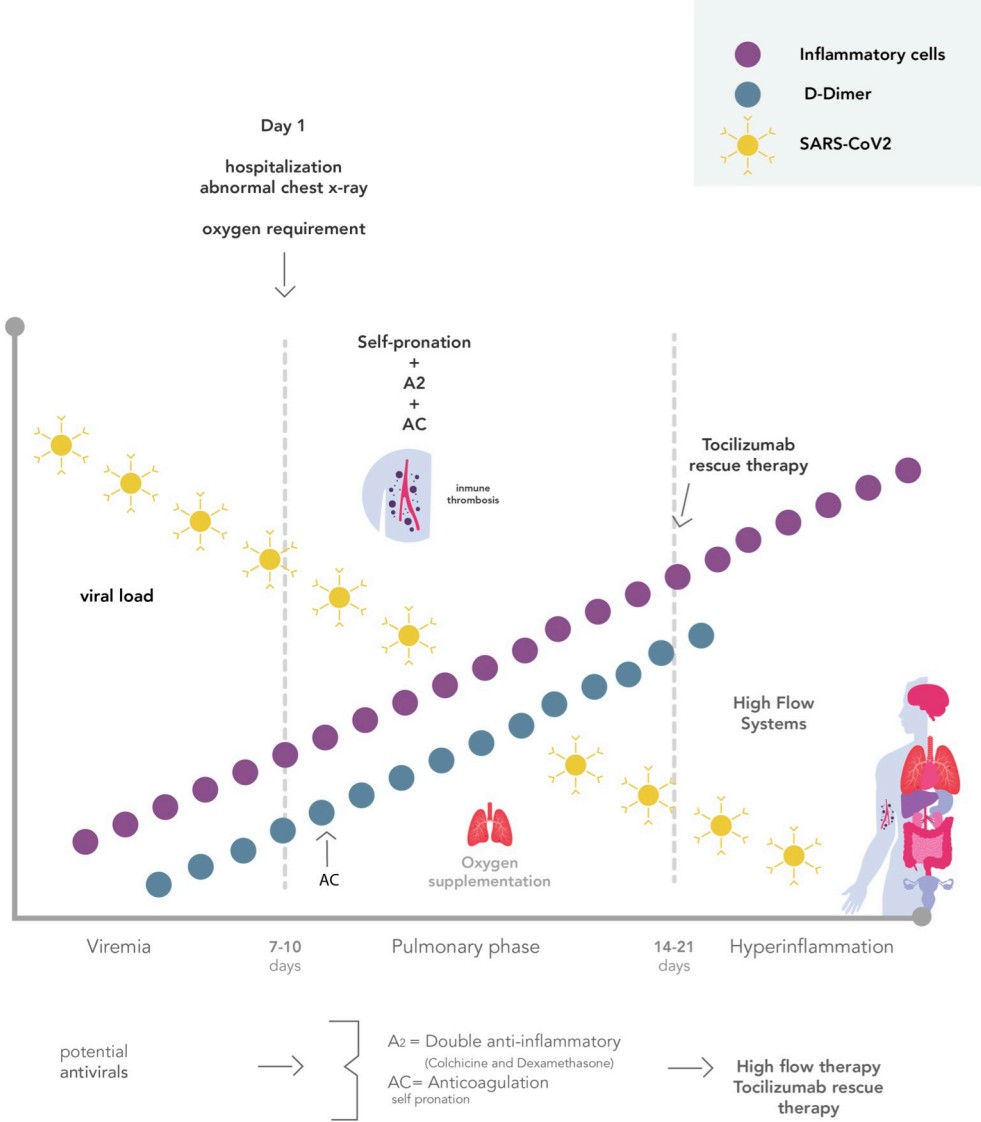

**Fig 1. Days following COVID-19 clinical phases.** *A2: double anti-inflammatory, AC: anticoagulation. MMA used in the hospital phases of COVID-19, illustrating timing is essential.

Hondureño de Seguridad Social (IHSS); 2. CEMESA Hospital in San Pedro Sula, Cortes, Honduras; 3. DeTar Healthcare systems; 4. Citizens Medical Center; 5. Del Campo Memorial Hospital in Texas, United States. Our hospital selection was based on the availability of ICU units and a clinical lab that had Ferritin, D-dimer, PCR, and PCT testing, and on the agreement of co-investigators to employ the MMA protocol. The hospitals in San Pedro Sula, Cortes, were selected because the virus spread rapidly across this city early in the pandemic. We did not invite other centers in Honduras, as they lack ICU units for COVI D 19 patients. The objective was to evaluate the outcome of the patients according to the treatment scheme received. The endpoints were evaluated using the variables ICU Length Of Stay (LOS) and mortality. Also, demographic data, co-morbidities, debut symptoms, acute phase reactants at the time of admission were analyzed. Patients were included if they were COVID-19 confirmed by RT-PCR, Early Warning scale (EWS) [14] greater than or equal to seven points and Quick

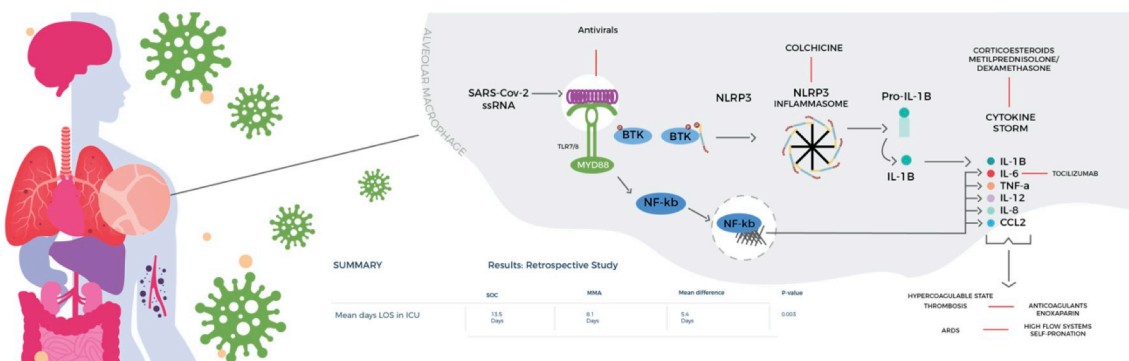

**Fig 2. Multi-mechanism Approach (MMA).** Includes widely available medications thought to target early immunomodulation, anticoagulation, and viral suppression to prevent catastrophic cytokine release syndrome and potential progression to respiratory failure, shock, and multi-organ dysfunction.

Sequential Organ Failure Assessment (Quick-SOFA) score [15] of one or higher at the time of admission to ICU; irrespective of gender and co-morbidities, patients under 18 years of age, pregnant, or breastfeeding were excluded.

Patients were grouped according to the treatment received, at the time of admission to ICU. The treatment schemes consisted of: 1. Standard of care was based on the recommendations provided by WHO as of March 13, 2020: which were Mechanical Ventilation with strategies for Adult Respiratory Distress Syndrome (ARDS), sedation, neuromuscular blockade, self- pronation, intravenous fluids for maintenance as well as antibiotics and vasopressor support if necessary (**Table 1**), and 2. The multi-mechanism approach plus standard of care, which targeted the different pathophysiological pathways likely involved in COVID-19 disease (**Fig 2**), adopted by the health authorities from the third week of April 2020, which consists of anticoagulation, immunomodulation (**Table 2**), and ventilatory strategies alternative to mechanical ventilation.

**Statistical methods.** Data were collected from clinical records at the time of the clinical outcome for each patient (discharge from ICU or death), and data were analyzed with the SPSS statistical application and R. Baseline characteristics of the participants were compared by the treatment scheme. Categorical variables are presented in number and percentage, and

**Table 2. Multi-Mechanism Approach (MMA): Medication regimen including dosage, frequency, route of administration, and duration of treatment.**

| Classification of Treatment | Medication | Dosage | Frequency | Route of Administration | Duration of Treatment |
|---|---|---|---|---|---|
| Anti-inflammatory | Methylprednisolone | 1–2 mg/kg/day | every 6 hours | Intravenous | 5–7 days |
| | Dexamethasone | Equivalent to Methylprednisolone | once a day | Intravenous | 5–7 days |
| | Colchicine | 1mg initial dose, 0.5mg subsequent doses (Honduras) | every 12 hours | Oral | 5 days |
| | | 1.2mg initial, 0.6mg subsequent doses (Texas) | every 12 hours | Oral | 5 days |
| Immunomodulator | Tocilizumab | 4–8 mg/kg/dose | once as needed | Intravenous | see footnote* |
| Anticoagulation | Low Molecular Weight Heparin | 1mg/kg | every 12 hours | subcutaneous | 14 days |

*the second dose was assessed at 24 hours according to the evolution, which consisted of the measurement of acute phase reactants and progression of mechanical ventilation parameters

were compared using the Chi-square test or Fisher's exact test as necessary. Numerical variables are presented with mean and standard deviation, and normally distributed variables were compared using Student's T with independent samples, whereas data with non-normal distributions were compared using the Mann Whitney test with independent samples. The two-tailed p-value less than 0.05 was considered statistically significant.

Death events and mean length of stay (LOS) in the ICU were modeled with all variables in univariate logistic regression models in SPSS to identify possible confounders to use in multivariate analysis as adjustments to the treatment effect.

Mean LOS in the ICU for each treatment group was first compared using an unadjusted Student's T-test, where the effect size is the difference in means. LOS was then modeled with multivariate linear regression to control for possible influences from other variables, including factors for hypertension, death, Quick SOFA of 2 to 3, mechanical ventilation, and gender, and the interval variable age. High flow was not included in the model because of co-linearity with mechanical flow (correlation of -1). The glm function in R was used for this model, which assumed LOS follows a distribution in the Gamma family. This generalized linear model fits the equation: $1/i = 0+1xi$.

To determine a best fit model, the full model was reduced to an optimal model with stepwise regression using the step AIC R function from the MASS package. The effect sizes in this regression analysis are the regression coefficients, but we can also estimate an adjusted difference of means from the regression equation. The reduced optimal model was used to predict an adjusted estimate of mean LOS in each treatment group with a mean age of 54.

Death events were modeled with multivariate logistic regression in R using the glm function, where the death event variable follows the binomial distribution, and the model consisted of the factors treatment group, hypertension, Quick-SOFA of 2 or 3, and gender, and the interval variable age. To determine a best fit model, the full model was reduced to an optimal model with stepwise regression using the stepAIC R function from the MASS package. The effect sizes for logistic regression are odds ratios.

**Ethics statement.** The institutional review boards approved the study as minimal risk for the retrospective character of the study and informed consent was not required (Citizens Medical Center, El Campo Memorial Hospital and Detar Health Care System, Victoria, Texas, USA and in San Pedro Sula, Honduras Centro Medico CEMESA and Instituto Hondureño de Seguridad Social).

## Results

### Baseline characteristics

A total of 86 patients were admitted to the ICU. Of these, 21 (24%) did not met inclusion criteria and 65 (76%) were included in the analysis. Of the 65 study participants, 41 (63%) were admitted to San Pedro Sula, Cortés Hospitals, and 24 (37%) from Victoria and El Campo, Texas, USA. There were 30 patients (46%) who received SC and 35 (54%) who received the MMA treatment. (**Fig 3**)

The mean age of participants was 54 years SD± 16.5 and range (18–86 years), with 46 (71%) males, 34 (52%) patients with more than one comorbidity, 33 (51%) patients with hypertension (the most frequent co-morbidity), 48 (74%) patients who survived, and 17 (26%) who died.

The standard care group had a mean age of 57.7 years SD ±16.3, with 9 (30%) patients that had two or more comorbidities. The most frequent debut symptom in this group was fever 19 (63%), with 11 patients (37%) presenting with a Q-SOFA ≥2. Mean lymphocytes in the SC group was 1065, with SD±755, and DD was 2.3, with SD ±2.6. Days of illness in the SC group at the time of admission were 7 (SD ± 4.5).

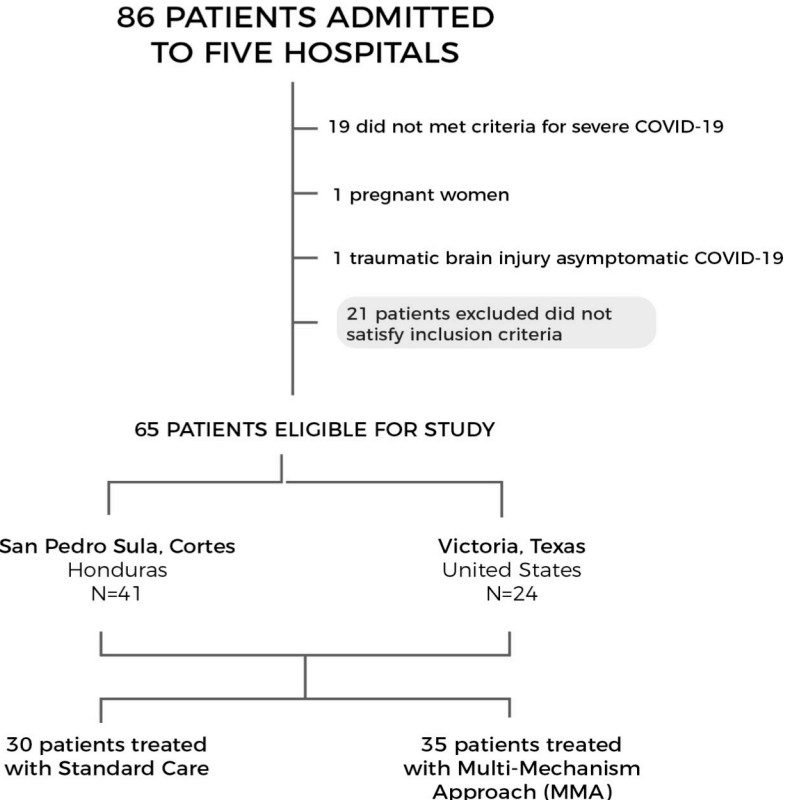

**Fig 3. Overview of participants includes in the MMA cohort.**

The MMA group had a mean age of 50.6 years SD ±15.9, with 13 (37%) who had two or more comorbidities. The predominant debut symptom was cough, with 18 patients (51%), and the average number of days of illness at the time of admission was 6.1. The mean DD in the MMA group was 0.89 (SD ± 0.94) and mean lymphocytes were 1147 (SD ±1032), Among baseline characteristics, the only statistically significant univariate difference between the treatment groups was the Q-SOFA Score (p-value = 0.02) (**Table 3**).

**Outcomes.** The primary outcome of the study was mortality and the secondary outcome was LOS in the ICU between treatment groups. Univariate mortality was found to be different between the SC (12 deaths / 30 patients) and MMA (5 deaths / 35 patients) treatment groups (p-value = 0.03) (**Table 4**). Treatment was not a significant factor for death events after adjustments for the factors hypertension, Quick SOFA of 2 or 3, and gender, and the interval variable age, in a multivariate logistic regression model of death events (**S3 Table**). This full regression model was reduced to an optimal model with stepwise regression, where the mortality difference failed to reach significance (p-value 0.14, regression coefficient 0.49). The optimal regression model for death events revealed that mechanical ventilation was the strongest predictor of death (p-value = 0.005, OR = 8.12) (**Table 5**)

Univariate analysis with Student's t-test revealed a significant difference in the mean LOS in the ICU between the MMA (8,06 days) and SC (14.43 days) groups (p-value = 0.001) (**Fig 4**). LOS in ICU was then modeled with multivariate linear regression to control for possible influences from other variables, including factors for hypertension, death, Quick SOFA of 2 to 3, mechanical ventilation, and gender, and the interval variable age (**S4 Table**). There were 49 males and 19 females in the study. The survival rate for both males and females was 74% (**S5**

**Table 3. Characteristics of the patients between treatment groups.**

| Characteristics | Standard Care N = 30 (%) | Multi-Mechanism Approach N = 35 (%) | Total N = 65 (%) | Range (min-max) | P value |
|---|---|---|---|---|---|
| **Age** | 57.7±16.3 | 50.6±15.9 | 53.9±16.4 | (18–86) | 0.08 |
| **Comorbidities** | | | | | |
| < 2 | 16 (53) | 18 (51) | 34 (52) | - | 0.85 |
| > 2 | 9 (30) | 13 (37) | 22 (34) | - | 0.36 |
| none | 5 (66) | 4 (11) | 9 (14) | - | 0.59 |
| Hypertension | 16 (53) | 17 (49) | 33 (51) | - | 0.70 |
| Obesity | 9 (30) | 13 (37) | 22 (34) | - | 0.54 |
| Diabetes | 7 (23) | 13 (37) | 20 (31) | - | 0.22 |
| Thyroid diseases | 3 (10) | 1 (3) | 4 (6) | - | 0.32 |
| Cardiovascular disease | 5 (17) | 1 (3) | 6 (9) | - | 0.08 |
| Asthma | 2 (7) | 2 (6) | 4 (6) | - | 1.0 |
| Cancer | 1 (3) | 2 (6) | 3 (5) | - | 1.0 |
| Other diseases | 8 (27) | 7 (20) | 15 (23) | - | 0.56 |
| **Gender** | | | | | 0.78 |
| Female | 8 (27) | 11 (31) | 19 (29) | - | - |
| Male | 22 (73) | 24 (69) | 46 (71) | - | - |
| **Severity scales** | | | | | |
| **EWS** | | | | | |
| ≥ 7 | 30 (100) | 35 (100) | 65 (100) | - | - |
| **Quick SOFA** | | | | | |
| **1** | 19 (63) | 31 (89) | 50 (77) | - | 0.02 |
| **2–3** | 11 (37) | 4 (11) | 15 (23) | - | |
| **Debut symptoms** | | | | | |
| Cough | 17 (57) | 18 (51) | 35 (54) | - | 0.80 |
| Fever | 19 (63) | 13 (37) | 32 (49) | - | 0.04 |
| Odynophagia | 1 (3) | 4 (11) | 5 (8) | - | 0.36 |
| Dyspnea | 2 (7) | 7 (20) | 9 (14) | - | 0.16 |
| Mild discomfort | 7 (44) | 9 (56) | 16 (25) | - | 1.0 |
| **Days from symptoms onset prior to presenting to the hospital** | 7.0±4.5 | 6.1±5.4 | 6.69±5.05 | (1–24) | 0.93 |
| **Laboratory findings at the admission to ICU** | | | | | |
| Lymphocyte count | 1065±755 | 1147±1032 | 1108± 907 | (171–6000) | 0.83 |
| Ferritin | 1530± 1195 | 1220±1190 | 1356±907 | (61–5343) | 0.27 |
| LDH | 529± 198 | 503±573 | 513± 463.52 | (134–3460) | 0.56 |
| D-Dimer | 2.3±2.6 | 0.89± 0.94 | 1.49± 1.97 | (0.1–8.37) | 0.04 |
| **Respiratory Support** | | | | | |
| Mechanical Ventilation | 14(47) | 2(6) | 16(25) | - | 0.0001 |
| High flow | 16(53) | 33(94) | 49(75) | - | 0.0001 |
| Required mechanical ventilation after High flow | 3(10) | 8(23) | 11(17) | - | 0.001 |
| Deaths after mechanical ventilation | 8(27) | 0 | 8(12) | - | 0.13 |

Sociodemographic characteristics, severity scales upon admission to the ICU, symptoms upon admission to the hospital, laboratory findings, and respiratory support needed.

**Table**). The coefficient for the treatment group was found to be significant in this model [p-value = 0.023) (**Table 4**). A reduced optimal model with the treatment factor and age was used to predict an adjusted estimate of mean LOS in each treatment group with a mean age of 54. In

**Table 4. Outcome: Mean LOS in the ICU for each treatment group was compared using an unadjusted Student's T-test for survivors and chi-squared test for event versus treatment group.**

| Outcome | MMA N = 35 | SOC N = 30 | Mean difference | P-value | 95% CI |
|---|---|---|---|---|---|
| LOS in ICU (mean days) | 7.3 | 14.2 | 6.9 days | 0.003 | 2.49–11.3 |
| Mortality (%) | 5(14) | 12(40) | - | 0.01 | - |

*LOS: length of stay, ICU: intensive care unit, MMA: multi-mechanism approach, SOC: standard of care, CI: confidence interval. Unadjusted analysis showing significant difference in mortality and length of stay reduction (between the survivors).

this age-adjusted model, the multi-mechanism approach treatment was associated with a mean LOS of 8.15 days, compared with a mean LOS under standard care of 13.55 days, with a mean reduction in length of stay in the ICU of 5.4 days (p-value: 0.03) (**Table 6**). There is a strong relationship between treatment and age as predictors of LOS, showing greater benefit in reducing LOS in ICU for older patients (up to ~9 days) (**Fig 5**).

## Discussion

Recent analysis of the early SARS-COV2 outbreak in Wuhan suggests that the number of infected people doubled every 2.3–3.3 days and each infected person went on to infect 5.7 additional people [16]. Since the virus is so contagious, the number of critically ill COVID-19 patients requiring admission rapidly exceeded bed capacity for hospitals in Wuhan in January 2020, which motivated the construction of two new hospitals with 2600 beds in 10 days [17]. In the summer of 2020, hospitals in the Honduras [18] and Texas [19] faced similar COVID-19 related capacity constraints.

The finding that MMA decreases the average intensive care unit (ICU) length of stay (LOS) by 5.4 days (adjusted for a mean age of 54 years) and up to 9 days (unadjusted for mean age) suggests that implementation of this treatment protocol could allow a healthcare system to manage 60% more COVID-19 patients with the same number of ICU beds as shown in Fig 5. At the time of this writing there are 7,028 COVID-19 patients in Texas hospitals [20]. Assuming adoption of MMA for treatment throughout Texas were to have had similar effects on length of stay, the decrease in the number of hospital beds required to treat that number of COVID-19 patients would be the functional equivalent of building new hospitals with 2800 beds. Although the mean length of stay was decreased by 5.4 days, it was also found that the improvement of length of stay improved to 9 days with advanced age as shown in Fig 5.

A 5.4 day ICU stay in the US for mechanically ventilated and non-ventilated patients has been to cost an estimated $37,258 and $28,852 per day respectively in 2020 [21]. There were 42,865 laboratory confirmed COVID-19 associated hospitalization reported by US sites between March 1, 2020 and August 1, 2020 [22]. As perhaps 20% of hospitalized COVID-19 patients need ICU beds [23], a 5.4 day decrease in ICU length of stay for that number of

**Table 5. Outcomes adjusted, a reduced optimal model was used to predict mortality adjusted by mechanical ventilation, hypertension and mean age.**

| Outcome Mortality Predictors | P-value | Regression Coefficient | OR |
|---|---|---|---|
| Mechanical Ventilation | 0.005 | 2.79 | 8.1 |
| Age | 0.07 | 1.8 | |
| Hypertension | 0.09 | 1.2 | 3.3 |
| Treatment Group MMA | 0.6 | -0.4 | |

*MMA (Multi-mechanism Approach), MMA failed to reach significance in terms of mortality

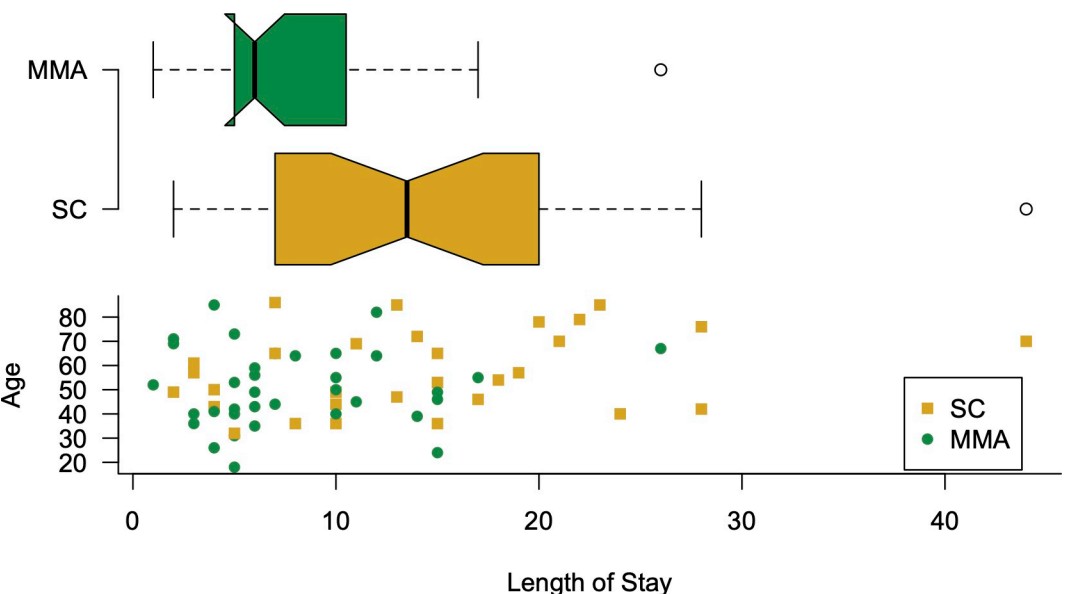

**Fig 4. Notched box plot, showing the difference in median LOS in ICU and scatterplot accounting age in y-axis.** *Visual indicator of the significant differences in median in LOS in ICU, the lower panel plots age and treatment were the only predictor variables.

COVID-19 patients could have resulted in a cost savings of between $247M and $318M, depending upon the percentage of patients needing mechanical ventilation.

Our finding that the MMA did not significantly reduce the overall adjusted ICU mortality rate was similar to results from the randomized trial of Remdesivir in severe COVID-19, where 28-day mortality in the Remdesivir-treated group was similar with placebo [24]. In both studies, the treatment was started late in the disease course [25]. The MMA treatment had a strong association with a decrease in ICU mortality rate when it was analyzed separately (without adjustment of key cofounders, p-value 0.001) 14%, compared to the standard treatment scheme with the mortality of 40%. The overall mortality rate among the participants was 26%, comparable to reports from Lombardy, Italy, and a multicenter study in China of 28% [26, 27].

Rapid outbreaks of COVID-19 cases are responsible for a total disruption of healthcare services. This was particularly true for early outbreaks, and selection bias in hospital admissions is one of the consequences of those circumstances. For example, hospital admissions for acute myocardial infarction and other acute cardiovascular diseases were dramatically reduced [28, 29]. We acknowledge that there could have been a selection bias in this study. However, the scope of our study addresses a treatment for critically-ill COVID-19 hospitalized patients in the ICU so therefore we do not believe that this selection bias affects the outcome of our study.

**Table 6. Age adjusted estimate for mean days in ICU for each treatment group.**

| Factor | SOC Mean days LOS in ICU | MMA Mean day LOS in ICU | Mean Difference | p-value |
|---|---|---|---|---|
| **Mean days adjusted by mean age 54 years** | 13.55 days | 8.15days | 5.4 days | 0.03 |

A reduced optimal model was used to predict an adjusted estimate of mean LOS in each treatment group with a mean age of 54, with significant reduction in adjusted LOS in ICU.

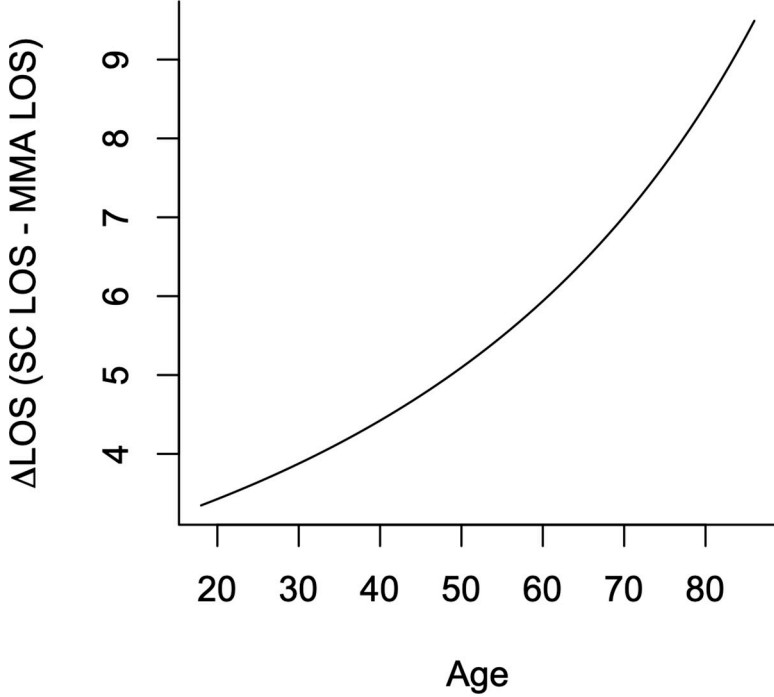

**Fig 5. Delta LOS in ICU.** *Based on the optimized model, which considers treatment and age as predictors of LOS, showing greater benefit in reducing LOS in older patients (up to 9 days).

## Limitations

This study included 5 different hospitals in two different countries, which accounts for the heterogenicity of the clinical characteristics of the patients and of the disease severity across the groups. The MMA treatment has small differences between the two collaborating regions, with different doses of colchicine and types of steroids used (dexamethasone with a glucorticoid effect in USA and methylprednisolone in Honduras), whereas, the use of a mineralocorticoid role has not yet been established as a primary pathophysiologic mechanism linked to mortality and complications in COVID-19. The heterogeneity of the study groups was recognized, observing that the non-exposed group had higher Q-SOFA, age, D-dimer levels, and the exposed group had more co-morbidities. In the unadjusted analysis, the use of MMA is significant in reducing mortality. However, after controlling key cofounders of mortality, including Q-SOFA score, age, gender, hypertension, and mechanical ventilation, there was no statistically significant difference in mortality between treatment groups. Further studies are needed to evaluate the optimal timing of MMA initiation based on the severity of disease stage. Many questions remain open, and the generalizability of the data must be considered in the relation of different epidemiological settings.

We examined mortality as a potential proxy for LOS and found no association between LOS and mortality. This is likely because of the small sample size in this study. Correcting for mortality (or even leaving all those patients who died out of the analysis) did not make a substantial difference in the LOS difference measured.

Age is a predictor of mortality in severe COVID-19 disease globally [30], but we did not identify an association between age and mortality in our study. The mean age in our study was 54 and our study population over 65 years of age was relatively small. The lack of association in our study between age and mortality is potentially explained by lower numbers of older

people reporting to the hospital during the early stages of the pandemic, likely due to national policies requesting people to stay home unless severely ill.

Hypertension was a key confounder in the analysis and it was used to adjust for the regression of mortality. It was statistically significant in the univariant analysis. Thus, we expect that potential acute cardiovascular and thrombotic events contributed to mortality in both groups, although those events were not captured by our analysis.

## Conclusion

The current study reveals that a distinct multi-mechanism approach for severe COVID-19 patients is associated with reduced mean LOS in the ICU of 5.4 days (adjusted for a mean age of 54 years) and up to 9 days (unadjusted for mean age), thus yielding further evidence to support the simultaneous use of ivermectin, colchicine, therapeutic anticoagulation and corticosteroids in a timely manner for better outcomes. The interventions are affordable and accessible to patients in both developing and developed countries. A mean reduction of 5.4 days (adjusted for a mean age of 54 years) and up to 9 days (unadjusted for mean age) in the ICU represents a substantial savings in healthcare costs and significant improvement to patient centered outcomes. Our study motivates future research in this area, including a randomized controlled trial of participants with moderate and severe COVID-19 to monitor the evolution of patient conditions alongside treatments. Further study is needed to explore the role of specific components of the MMA treatment in reducing LOS in ICU, and larger studies may reveal a potential effect of MMA on mortality.

## Supporting information

**S1 File.**
(DOCX)

**S1 Dataset.**
(CSV)

## Acknowledgments

We wish to thank Marco Tulio Medina, MD (Universidad Nacional Autónoma de Honduras) for his expert advice, and to Mr. Daniel Antonio Fortin, who brought members of this group together to work on the development of this manuscript. Special thanks go to the Horwath Central America team.

In loving memory of Luis Enamorado, MD from Honduras and Christopher Cornish, RT from South Texas who were frontline healthcare providers in the ICU but succumbed in the battle against COVID 19.

## Author Contributions

**Conceptualization:** Fernando Valerio Pascua, Oscar Diaz, Rina Medina, Brian Contreras, Jeff Mistroff, Daniel Espinosa, Anupamjeet Sekhon, Diego Paz Handal, Estela Pineda, Miguel Vargas Pineda, Hector Pineda, Maribel Diaz, Anita S. Lewis, Heike Hesse, Miriams T. Castro Lainez, Mark L. Stevens, Miguel Sierra- Hoffman, Sidney C. Ontai, Vincent VanBuren.

**Data curation:** Fernando Valerio Pascua, Oscar Diaz, Rina Medina, Brian Contreras, Jeff Mistroff, Daniel Espinosa, Anupamjeet Sekhon, Diego Paz Handal, Estela Pineda, Miguel Vargas Pineda, Hector Pineda, Maribel Diaz, Anita S. Lewis, Heike Hesse, Miriams T. Castro Lainez, Mark L. Stevens, Miguel Sierra- Hoffman, Sidney C. Ontai, Vincent VanBuren.

**Formal analysis:** Fernando Valerio Pascua, Oscar Diaz, Rina Medina, Brian Contreras, Jeff Mistroff, Daniel Espinosa, Anupamjeet Sekhon, Diego Paz Handal, Estela Pineda, Miguel Vargas Pineda, Hector Pineda, Maribel Diaz, Anita S. Lewis, Heike Hesse, Miriams T. Castro Lainez, Mark L. Stevens, Miguel Sierra- Hoffman, Sidney C. Ontai, Vincent VanBuren.

**Funding acquisition:** Oscar Diaz.

**Investigation:** Fernando Valerio Pascua, Oscar Diaz, Rina Medina, Brian Contreras, Jeff Mistroff, Daniel Espinosa, Anupamjeet Sekhon, Diego Paz Handal, Estela Pineda, Miguel Vargas Pineda, Hector Pineda, Maribel Diaz, Anita S. Lewis, Heike Hesse, Miriams T. Castro Lainez, Mark L. Stevens, Miguel Sierra- Hoffman, Sidney C. Ontai, Vincent VanBuren.

**Methodology:** Fernando Valerio Pascua, Oscar Diaz, Rina Medina, Brian Contreras, Jeff Mistroff, Daniel Espinosa, Anupamjeet Sekhon, Diego Paz Handal, Estela Pineda, Miguel Vargas Pineda, Hector Pineda, Maribel Diaz, Anita S. Lewis, Heike Hesse, Miriams T. Castro Lainez, Mark L. Stevens, Miguel Sierra- Hoffman, Sidney C. Ontai, Vincent VanBuren.

**Project administration:** Fernando Valerio Pascua, Oscar Diaz, Rina Medina, Brian Contreras, Jeff Mistroff, Daniel Espinosa, Anupamjeet Sekhon, Diego Paz Handal, Estela Pineda, Miguel Vargas Pineda, Hector Pineda, Maribel Diaz, Anita S. Lewis, Heike Hesse, Miriams T. Castro Lainez, Mark L. Stevens, Miguel Sierra- Hoffman, Sidney C. Ontai, Vincent VanBuren.

**Resources:** Fernando Valerio Pascua, Oscar Diaz, Rina Medina, Brian Contreras, Jeff Mistroff, Daniel Espinosa, Anupamjeet Sekhon, Diego Paz Handal, Estela Pineda, Miguel Vargas Pineda, Hector Pineda, Maribel Diaz, Anita S. Lewis, Heike Hesse, Miriams T. Castro Lainez, Mark L. Stevens, Miguel Sierra- Hoffman, Sidney C. Ontai, Vincent VanBuren.

**Software:** Fernando Valerio Pascua, Oscar Diaz, Rina Medina, Brian Contreras, Jeff Mistroff, Daniel Espinosa, Anupamjeet Sekhon, Diego Paz Handal, Estela Pineda, Miguel Vargas Pineda, Hector Pineda, Maribel Diaz, Anita S. Lewis, Heike Hesse, Miriams T. Castro Lainez, Mark L. Stevens, Miguel Sierra- Hoffman, Sidney C. Ontai, Vincent VanBuren.

**Supervision:** Fernando Valerio Pascua, Oscar Diaz, Rina Medina, Brian Contreras, Jeff Mistroff, Daniel Espinosa, Anupamjeet Sekhon, Diego Paz Handal, Estela Pineda, Miguel Vargas Pineda, Hector Pineda, Maribel Diaz, Anita S. Lewis, Heike Hesse, Miriams T. Castro Lainez, Mark L. Stevens, Miguel Sierra- Hoffman, Sidney C. Ontai, Vincent VanBuren.

**Validation:** Fernando Valerio Pascua, Oscar Diaz, Rina Medina, Brian Contreras, Jeff Mistroff, Daniel Espinosa, Anupamjeet Sekhon, Diego Paz Handal, Estela Pineda, Miguel Vargas Pineda, Hector Pineda, Maribel Diaz, Anita S. Lewis, Heike Hesse, Miriams T. Castro Lainez, Mark L. Stevens, Miguel Sierra- Hoffman, Sidney C. Ontai, Vincent VanBuren.

**Visualization:** Fernando Valerio Pascua, Oscar Diaz, Rina Medina, Brian Contreras, Jeff Mistroff, Daniel Espinosa, Anupamjeet Sekhon, Diego Paz Handal, Estela Pineda, Miguel Vargas Pineda, Hector Pineda, Maribel Diaz, Anita S. Lewis, Heike Hesse, Miriams T. Castro Lainez, Mark L. Stevens, Miguel Sierra- Hoffman, Sidney C. Ontai, Vincent VanBuren.

**Writing – original draft:** Fernando Valerio Pascua, Oscar Diaz, Rina Medina, Brian Contreras, Jeff Mistroff, Daniel Espinosa, Anupamjeet Sekhon, Diego Paz Handal, Estela Pineda, Miguel Vargas Pineda, Hector Pineda, Maribel Diaz, Anita S. Lewis, Heike Hesse, Miriams T. Castro Lainez, Mark L. Stevens, Miguel Sierra- Hoffman, Sidney C. Ontai, Vincent VanBuren.

**Writing – review & editing:** Fernando Valerio Pascua, Oscar Diaz, Rina Medina, Brian Contreras, Jeff Mistroff, Daniel Espinosa, Anupamjeet Sekhon, Diego Paz Handal, Estela Pineda, Miguel Vargas Pineda, Hector Pineda, Maribel Diaz, Anita S. Lewis, Heike Hesse, Miriams T. Castro Lainez, Mark L. Stevens, Miguel Sierra- Hoffman, Sidney C. Ontai, Vincent VanBuren.

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
