## [Decision Letter · Decision Letter 0]

30 Sep 2020

PONE-D-20-29696

A multi-mechanism approach reduces length of stay in the ICU for severe COVID-19 patients

PLOS ONE

Dear Dr. VanBuren,

Thank you for submitting your manuscript to PLOS ONE. After careful consideration, we feel that it has merit but does not fully meet PLOS ONE’s publication criteria as it currently stands. Therefore, we invite you to submit a revised version of the manuscript that addresses the points raised during the review process.

Both reviewers found merits in this work, but their also raised some important issues that need to be addressed. I hope that the authors can effectively respond to their comments in the revision.

We look forward to receiving your revised manuscript.

Kind regards,

Yu Ru Kou, PhD

Academic Editor

PLOS ONE

Journal Requirements:

2. Please provide additional details regarding participant consent. In your ethics statement in the Methods section and in the online submission information, please clarify whether the need for consent was specifically waived by the ethics committee.

3. Please provide citations for all scales and scoring tools used in this study.

4. Please include the date(s) on which you accessed the databases or records to obtain the data used in your study.

5. Thank you for including your ethics statement: 'The institution's ethics committee approved the study and informed consent was not required.'

7. We note you have included tables to which you do not refer in the text of your manuscript. Please ensure that you refer to Tables 5 and 6 in your text; if accepted, production will need this reference to link the reader to the Table.

8. Your ethics statement should only appear in the Methods section of your manuscript. If your ethics statement is written in any section besides the Methods, please delete it from any other section.

Reviewers' comments:

Reviewer's Responses to Questions

**Comments to the Author**

1. Is the manuscript technically sound, and do the data support the conclusions?

Reviewer #1: Yes

Reviewer #2: Yes

2. Has the statistical analysis been performed appropriately and rigorously? 

Reviewer #1: Yes

Reviewer #2: Yes

3. Have the authors made all data underlying the findings in their manuscript fully available?

Reviewer #1: No

Reviewer #2: No

4. Is the manuscript presented in an intelligible fashion and written in standard English?

Reviewer #1: Yes

Reviewer #2: Yes

5. Review Comments to the Author

Reviewer #1: The authors present original data on the use of a multiple mechanism therapeutic approach (MMA) on patients with severe COVID-19 hospitalized in ICU. The report that the MMA was assciated with a decreases in average ICU length of stay, thereby causing a relevant unload of the hospital workflow arounf COVID-19, which was a critical issue during the first outbreak.

Specific comments:

- the rapid outbreak of COVID-19 cases especially during the first breakout was responsible for a total derangement of healthcare services. One of the consequences was the generation of a strong selection bias on hospital admissions. In fact, hospital admission for Acute Myocardial Infarction and other acute cardiovascular diseases were dramaticaly reduced (Reduction of hospitalizations for myocardial infarction in Italy in the COVID-19 era. Eur Heart J. 2020;41(22):2083-2088. doi: 10.1093/eurheartj/ehaa409. - COVID-19 pandemic and admission rates for and management of acute coronary syndromes in England. Lancet. 2020;396(10248):381-389. doi: 10.1016/S0140-6736(20)31356-8.). Similarly, a selection bias was also suggested for COVID-19 patients, whereas the most severe cases probably did not make it to the hospital, which might then have caused an underestiation of death. Please comment on this issue;

- the authors report IOT with mechanical ventilation being the only independent predictor of in-hospital death in this cohort. A recent analysis incuding over 75000 COVID-19 patients, of which 4344 were under intensive care found age, cardiovascular risk factors or comorbdities and CV complications were independent predictors of in-hospital death (Impact of cardiovascular risk profile on COVID-19 outcome. A meta-analysis. PLoS ONE 2020; 15(8): e0237131. https://doi.org/10.1371/journal.pone.0237131). Did the authors find similar results in their cohort? please discuss this aspect in the manuscript;

- in this regard, the auhtors report "a strong relationship between treatment and age as predictors of LOS, showing greater benefit in reducing LOS in ICU for older patients". In light of this finding, how do the authors explain the lack of association between age and death? Might it be related with the limited sample size?

- the authors state that "The database analyzed during the current study are available from the corresponding author on request.". HOwever, this doesn't comply to journal policies on data sharing. Please refer to authors' guidelines;

Reviewer #2: The authors assessed the impact of a "multiple mechanism therapeutic approach" (MMA) on the clinical management of patients with severe COVID-19. They found that the "MMA" approach was assciated with a decrease in length of stay in the ICU.

Comments:

- please, describe the criteria for selection of study centers. Please, also report how many centers were invited and the percentage of participating centers from those invited;

- how were clinical endpoints reported? do the authors have information on thrombotic events?

- despite many efforts, clinical information on female patients is still underrepresented compared to males. This issue has been evan larger with COVID-19. (Sabatino J. et al. Women's perspective on the COVID-19 pandemic: Walking into a post-peak phase. Int J Cardiol. 2020:S0167-5273(20)33552-X. doi: 10.1016/j.ijcard.2020.08.025.). Could the authors please report their results stratified by gender (e.g. in a summary table) and comment about eventual differences?

- lenght of stay is an obvious proxy of mortality, how did the authors managed the shorter LOS for early deaths? was any correction applied?

- a recent meta-analysis including over 4000 COVID-19 patients under intensive care identified age as an independent predictor of in-hospital death (Sabatino J. et al. Impact of cardiovascular risk profile on COVID-19 outcome. A meta-analysis. PLoS One. 2020;15(8):e0237131. doi: 10.1371/journal.pone.0237131.). How do the authors explain the lack of association in their cohort?

- Did the authors find any association between cardiovascular comorbidities and in-hospital death?

6. PLOS authors have the option to publish the peer review history of their article (what does this mean?). If published, this will include your full peer review and any attached files.

Reviewer #1: No

Reviewer #2: No

---

## [Author Response · Author response to Decision Letter 0]

10 Dec 2020

Reviewer #1: The authors present original data on the use of a multiple mechanism therapeutic approach (MMA) on patients with severe COVID-19 hospitalized in ICU. The report that the MMA was assciated with a decreases in average ICU length of stay, thereby causing a relevant unload of the hospital workflow arounf COVID-19, which was a critical issue during the first outbreak.

Specific comments:

the rapid outbreak of COVID-19 cases especially during the first breakout was responsible for a total derangement of healthcare services. One of the consequences was the generation of a strong selection bias on hospital admissions. In fact, hospital admission for Acute Myocardial Infarction and other acute cardiovascular diseases were dramaticaly reduced (Reduction of hospitalizations for myocardial infarction in Italy in the COVID-19 era. Eur Heart J. 2020;41(22):2083-2088. doi: 10.1093/eurheartj/ehaa409. - COVID-19 pandemic and admission rates for and management of acute coronary syndromes in England. Lancet. 2020;396(10248):381-389. doi: 10.1016/S0140-6736(20)31356-8.). Similarly, a selection bias was also suggested for COVID-19 patients, whereas the most severe cases probably did not make it to the hospital, which might then have caused an underestiation of death. Please comment on this issue;

RESPONSE: Yes we agree and we will make the following comments and cite the references in the manuscript:

Rapid outbreaks of COVID-19 cases are responsible for a total disruption of healthcare services. This was particularly true for early outbreaks, and selection bias in hospital admissions is one of the consequences of those circumstances. For example, hospital admissions for acute myocardial infarction and other acute cardiovascular diseases were dramatically reduced [citations]. We acknowledge that there could have been a selection bias in this study. However, the scope of our study addresses a treatment for critically-ill COVID-19 hospitalized patients in the ICU so therefore we do not believe that this selection bias affects the outcome of our study.

- the authors report IOT with mechanical ventilation being the only independent predictor of in-hospital death in this cohort. A recent analysis incuding over 75000 COVID-19 patients, of which 4344 were under intensive care found age, cardiovascular risk factors or comorbdities and CV complications were independent predictors of in-hospital death (Impact of cardiovascular risk profile on COVID-19 outcome. A meta-analysis. PLoS ONE 2020; 15(8): e0237131. https://doi.org/10.1371/journal.pone.0237131). Did the authors find similar results in their cohort? please discuss this aspect in the manuscript;

RESPONSE:Mechanical ventilation was the strongest predictor of mortality in this cohort. However, age and cardiovascular complications have been identified as confounding factors [citation] and were used to adjust the logistic regression model for mortality.

- in this regard, the auhtors report "a strong relationship between treatment and age as predictors of LOS, showing greater benefit in reducing LOS in ICU for older patients". In light of this finding, how do the authors explain the lack of association between age and death? Might it be related with the limited sample size?

RESPONSE: The lack of association between age and death is likely due to the limited sample size in this study. We amend the manuscript as described in another response below.

the authors state that "The database analyzed during the current study are available from the corresponding author on request.". HOwever, this doesn't comply to journal policies on data sharing. Please refer to authors' guidelines;

RESPONSE: We will make the data available as a supplementary file.

Reviewer #2: The authors assessed the impact of a "multiple mechanism therapeutic approach" (MMA) on the clinical management of patients with severe COVID-19. They found that the "MMA" approach was assciated with a decrease in length of stay in the ICU.

Comments:

please, describe the criteria for selection of study centers. Please, also report how many centers were invited and the percentage of participating centers from those invited;

Our hospital selection was based on the availability of ICU units and a clinical lab that had Ferritin, D-dimer, PCR, and PCT testing. The hospitals in San Pedro Sula, Cortes, were selected because this the virus spread rapidly across this city. We did not invite other centers in Honduras, as they lack of materials or ICU units for COVID 19 patients. 

how were clinical endpoints reported? do the authors have information on thrombotic events?

RESPONSE:Endpoints for patients were either death or discharge from the hospital. Information on thrombotic events were not captured for this study.

- despite many efforts, clinical information on female patients is still underrepresented compared to males. This issue has been evan larger with COVID-19. (Sabatino J. et al. Women's perspective on the COVID-19 pandemic: Walking into a post-peak phase. Int J Cardiol. 2020:S0167-5273(20)33552-X. doi: 10.1016/j.ijcard.2020.08.025.). Could the authors please report their results stratified by gender (e.g. in a summary table) and comment about eventual differences?

RESPONSE: A table that stratifies results by gender has been added to the manuscript. 

There were 49 males and 19 females in the study. The survival rate for both males and females was 74% [S5 Table]. 

Given the the relatively small number of females in this study, we are not confident in providing further analysis of stratified data.

lenght of stay is an obvious proxy of mortality, how did the authors managed the shorter LOS for early deaths? was any correction applied?

RESPONSE: Limitations:

We examined mortality as a potential proxy for LOS and found no association between LOS and mortality. This is likely because of the small sample size in this study. Correcting for mortality (or even leaving all those patients who died out of the analysis) did not make a substantial difference in the LOS difference measured.

a recent meta-analysis including over 4000 COVID-19 patients under intensive care identified age as an independent predictor of in-hospital death (Sabatino J. et al. Impact of cardiovascular risk profile on COVID-19 outcome. A meta-analysis. PLoS One. 2020;15(8):e0237131. doi: 10.1371/journal.pone.0237131.). How do the authors explain the lack of association in their cohort?

RESPONSE: yes we agree with the findings that age is an independent predictor of in-hospital death and will addend the manuscript as follows:

Limitations:

Age is a predictor of mortality in severe COVID-19 disease globally [cite paper above], but we did not identify an association between age and mortality in our study.The mean age in our study was 54 and our study population over 65 years of age was relatively small. The lack of association in our study between age and mortality is potentially explained by lower numbers of older people reporting to the hospital during the early stages of the pandemic, likely due to national policies requesting people to stay home unless severely ill.

Did the authors find any association between cardiovascular comorbidities and in-hospital death?

RESPONSE: Yes we found the association between cardiovascular comorbidities and in-hospital death was found. We will amend the manuscript as follows:

Limitations:

Hypertension was a key confounder in the analysis and it was used to adjust for the regression of mortality. It was statistically significant in the univariant analysis. Thus, we expect that potential acute cardiovascular and thrombotic events contributed to mortality in both groups, although those events were not captured by our analysis.

---

## [Decision Letter · Decision Letter 1]

21 Dec 2020

A multi-mechanism approach reduces length of stay in the ICU for severe COVID-19 patients

PONE-D-20-29696R1

Dear Dr. VanBuren,

We’re pleased to inform you that your manuscript has been judged scientifically suitable for publication and will be formally accepted for publication once it meets all outstanding technical requirements.

Kind regards,

Yu Ru Kou, PhD

Academic Editor

PLOS ONE

Additional Editor Comments (optional):

The original submission was rated as "minor revision" by reviewer #1 who declined to re-evaluate the R1 version of revised manuscript. I have read through the responses from the authors to reviewer's comments. In my view, the authors have adequately revised their manuscript.

Reviewers' comments:

Reviewer's Responses to Questions

**Comments to the Author**

1. If the authors have adequately addressed your comments raised in a previous round of review and you feel that this manuscript is now acceptable for publication, you may indicate that here to bypass the “Comments to the Author” section, enter your conflict of interest statement in the “Confidential to Editor” section, and submit your "Accept" recommendation.

Reviewer #2: All comments have been addressed

2. Is the manuscript technically sound, and do the data support the conclusions?

Reviewer #2: Yes

3. Has the statistical analysis been performed appropriately and rigorously? 

Reviewer #2: Yes

4. Have the authors made all data underlying the findings in their manuscript fully available?

Reviewer #2: Yes

5. Is the manuscript presented in an intelligible fashion and written in standard English?

Reviewer #2: Yes

6. Review Comments to the Author

Reviewer #2: The authors have revised their manuscript. All comments have been addressed. I have no further comments.

7. PLOS authors have the option to publish the peer review history of their article (what does this mean?). If published, this will include your full peer review and any attached files.

Reviewer #2: No

---

## [Editor Report · Acceptance letter]

28 Dec 2020

PONE-D-20-29696R1 

A multi-mechanism approach reduces length of stay in the ICU for severe COVID-19 patients 

Dear Dr. VanBuren:

I'm pleased to inform you that your manuscript has been deemed suitable for publication in PLOS ONE. Congratulations! Your manuscript is now with our production department. 

Kind regards, 

on behalf of

Dr. Yu Ru Kou 

Academic Editor

PLOS ONE